# l-Malate (−2) Protonation State is Required for Efficient Decarboxylation to l-Lactate by the Malolactic Enzyme of *Oenococcus oeni*

**DOI:** 10.3390/molecules25153431

**Published:** 2020-07-28

**Authors:** Waldo Acevedo, Pablo Cañón, Felipe Gómez-Alvear, Jaime Huerta, Daniel Aguayo, Eduardo Agosin

**Affiliations:** 1Institute of Chemistry, Facultad de Ciencias, Pontificia Universidad Católica de Valparaíso, Valparaíso 2373223, Chile; waldo.acevedo@pucv.cl; 2Department of Chemical and Bioprocess Engineering, School of Engineering, Pontificia Universidad Católica de Chile, Santiago 7810000, Chile; pmcanon@uc.cl; 3Center for Bioinformatics and Integrative Biology (CBIB), Facultad de Ciencias Biológicas, Universidad Andres Bello, Santiago 8370146, Chile; fgomeza17@gmail.com (F.G.-A.); j.huerta.bq@gmail.com (J.H.); daniel.aguayo@unab.cl (D.A.); 4Interdisciplinary Center for Neuroscience of Valparaíso, Faculty of Science, University of Valparaíso, Valparaíso 2340000, Chile

**Keywords:** malolactic enzyme, reaction mechanism, docking, molecular dynamics, isothermal titration calorimetry

## Abstract

Malolactic fermentation (MLF) is responsible for the decarboxylation of l-malic into lactic acid in most red wines and some white wines. It reduces the acidity of wine, improves flavor complexity and microbiological stability. Despite its industrial interest, the MLF mechanism is not fully understood. The objective of this study was to provide new insights into the role of pH on the binding of malic acid to the malolactic enzyme (MLE) of *Oenococcus oeni.* To this end, sequence similarity networks and phylogenetic analysis were used to generate an MLE homology model, which was further refined by molecular dynamics simulations. The resulting model, together with quantum polarized ligand docking (QPLD), was used to describe the MLE binding pocket and pose of l-malic acid (MAL) and its l-malate (−1) and (−2) protonation states (MAL^−^ and MAL^2−^, respectively). MAL^2−^ has the lowest ∆G_binding_, followed by MAL^−^ and MAL, with values of −23.8, −19.6, and −14.6 kJ/mol, respectively, consistent with those obtained by isothermal calorimetry thermodynamic (ITC) assays. Furthermore, molecular dynamics and MM/GBSA results suggest that only MAL^2−^ displays an extended open conformation at the binding pocket, satisfying the geometrical requirements for Mn^2+^ coordination, a critical component of MLE activity. These results are consistent with the intracellular pH conditions of *O. oeni* cells—ranging from pH 5.8 to 6.1—where the enzymatic decarboxylation of malate occurs.

## 1. Introduction

Most red wines, as well as some white and sparkling wines, are produced by two sequential fermentations: first, yeast alcoholic fermentation (AF) transforms grape must into wine; then, a secondary fermentation, called malolactic fermentation (MLF), is carried out by lactic acid bacteria (LAB). Contrary to AF, a mandatory process of winemaking, MLF is optional and depends on the desired wine style. MLF reduces the acidity of wine and improves flavor complexity and microbiological stability. MLF involves the NAD^+^- and manganese-dependent decarboxylation of l-malate to l-lactate and CO_2_ [1,2]. Due to its monocarboxylic nature, lactic acid imparts a more elegant and round taste to the wine, as opposed to the astringent taste of the dicarboxylic, malic acid [3,4,5]. *Oenococcus oeni* is the main lactic acid bacterium involved in MLF, thanks to its ability to grow under the harsh conditions of wine, such as high ethanol content (>13% *v/v*), low pH (<3.5), and high sulphite concentration (>50 ppm) [6,7,8,9,10]. MLF is considered one of the most difficult processes to manage during winemaking, because it is often delayed or simply not fully achieved.

Among the issues of malolactic fermentation [11,12,13], the mechanism involved in the enzymatic reaction is perhaps the most unpredictable to ensure efficient and reproducible malate decarboxylation. Three decarboxylation pathways have been proposed so far, as shown in Figure 1 [14]. The first one considers that a malic enzyme (ME), followed by a l-lactate dehydrogenase, transforms malate to pyruvate, and then to lactate [15]. A second mechanism postulates a three-step reaction, mediated by a l-malate dehydrogenase (MDH), an oxaloacetate decarboxylase and a l-lactate dehydrogenase, respectively [16]. Finally, the third mechanism consists in the direct conversion of l-malate into l-lactate [17], through a reaction carried out in the presence of NAD^+^ and Mn^2+^, and where neither reduction of NAD^+^ nor detection of free reaction intermediates occurs [14,18]. This mechanism is conducted by a protein different from a previously described malic enzyme, the “malolactic enzyme” (MLE) [19].

The malolactic enzyme has been purified from several LAB, e.g., *Lactobacillus spp*., *Lactococcus lactis,* and *O. oeni* [7,17,20,21]. In all cases, the molecular mass of the MLE subunits range from 60 to 70 kDa, and the active form of the protein has been described either as a dimer or a tetramer of identical subunits [22,23,24,25], although all subunits have an independent active site. Interestingly, MLE active sites have binding sites with different structural arrangements of the amino acids Asp, Lys, Ser, and Tyr that, altogether, satisfy the coordination of divalent cation and cofactor positioning [15]. It is noteworthy that all described MLEs catalyze the same reaction.

Among LAB, *O. oeni* is the best adapted to the harsh conditions of wine. It is capable of carrying out spontaneous fermentation even at pH 3.2, a condition that could be found in some wines [8]. The objective of this study was to provide new insights on the reaction mechanism of the malolactic enzyme (MLE) of *O. oeni*, responsible for the transformation of l-malate into l-lactate in wine. To this end, we first expressed the MLE gene of *O. oeni* in *Escherichia coli* BL21. After purification of the protein, we measured the affinity of MLE for malic acid under several pH conditions by isothermal titration calorimetry. Then, we generated a MLE homology model, based on sequence similarity networking and phylogenetic analysis, in order to describe the MLE-malic acid molecular interaction at an atomic level, using molecular dynamics simulations. Finally, we explored the effect of pH on l-malate binding free energies and identified possible residues involved into malic acid binding by means of quantum polarized ligand docking and MM/GBSA calculations. To the best of our knowledge, this is the first study that explores the three-dimensional structure of the malolactic enzyme and its interaction with malic acid at the binding site, the first step of the reaction.

## 2. Results

### 2.1. Calorimetric Determination of Malic acid Binding Energies to Malolactic Enzyme

Malolactic fermentation in wine is usually carried out at a pH range between 3.2 and 3.5, allowing a small rise in pH as the malic acid is converted to lactic acid. Figure 2 illustrates the 2D-structure of l-malic acid (MAL) and its l-malate (−1) and (−2) protonation forms (MAL^−^ and MAL^2−^, respectively). Isothermal calorimetry thermodynamic (ITC) data for malic acid interaction with MLE shows dissociation constant (K_d_, Table 1) values in the micromolar range (Figure 3). MAL^2−^ has a lower ΔG value than MAL^1−^, suggesting this form as the most probable protonation state for malic acid at the binding site of MLE.

### 2.2. Sequence Similarity Networks of Malolactic Enzyme Family

To apprehend the impact of pH on *O. oeni* MLE activity, we performed an *in-silico* analysis by comparing its sequence with other MLF-related proteins, including malic enzyme and malate dehydrogenase. For this purpose, we generated a sequence similarity network (SSN), where nodes correspond to homologous proteins to MLE, i.e., those containing at least 70% of sequence identity; and where connections allow to rapidly compute and visualize groups of proteins based on all-against-all sequence comparisons (Figure 4). The group that contains *O. oeni* MLE and its closest homologues from NCBI’s non-redundant (nr) protein database are referenced as malate dehydrogenases, malic enzymes, and malolactic enzymes. Interestingly, most sequences of this group corresponding to malolactic enzymes and malic enzymes, therefore crystal structures of malic enzymes are the most adequate structural templates to model MLE as there is no structural data available for malolactic enzymes.

### 2.3. Phylogenetics of Malolactic Enzyme Family

Identification of a set of orthologs is a prerequisite for a robust genetic analysis of the evolution of a group of organisms [26]. We carried out multiple sequence alignment using CLUSTAL OMEGA to study the evolutionary relationships between different lactic bacteria in relation to the malolactic enzyme [27]. Most of the sequences homologous to the malolactic enzyme of *O. oeni*, correspond to proteins whose function has been assigned as malic enzymes by automatic annotation. However, some sequences have been experimentally reported as enzymes with malolactic activity. The latter were labelled with the abbreviation “MLE” below the name of the species, in the phylogenetic tree (Figure 5).

As illustrated in Figure 5, *O. oeni* is part of a monophyletic group, together with *Streptococcus spp, Lactococcus spp,* and *Enterococcus spp*. However, the evolution of the malic enzyme would be basal in *O. oeni* with respect to the rest of this clade. It is noteworthy that, within the clade representing the *Streptococcus* branch, the group of *Lactococcus* and *Enterococcus* are represented as sister groups of recent evolution. On the other hand, the branches of *Lactobacillus, Pediococcus,* and *Leuconostoc* constitute a paraphyletic group of basal character with respect to *Oenococcus* and *Streptococcus-Lactococcus-Enterococcus*; and they have a previous evolutionary origin.

### 2.4. Structural Modeling of the Malolactic Enzyme

The active site of the chain A of the malic enzyme from pigeon liver (PDB entry 1GQ2), the first malic enzyme described [19,28], was selected after SSN analysis as the best structural template for comparative modeling of MLE. The sequence alignment of both structures showed a sequence identity of 35.9% and a coverage of 98% against MLE (Figure 6).

The crystal structure of the A chain contains an oxalate ion in the binding site, and requires Mn^2+^ and NADP^+^ as cofactors. Nevertheless, supported by the highly conserved structure of the active site in both proteins, we confirmed that the putative active binding site could correctly locate malate, after replacing the former cofactors with NAD^+^ and Mn^2+^, and oxalate with malate, using SiteMap of Maestro suite (data not shown). It is worthy to mention that we also employed the malic enzyme from *E. coli* (PDB entry 6AGS) (Figure 7) for these purposes; though only as secondary scaffold, because no experimental data is available for this protein crystal.

Figure 7A shows the MLE homology model we obtained from the abovementioned templates and sequence alignments. This monomeric model was submitted to 200 ns simulation, reaching structural stability after 50 ns, by the structural rearrangement of the carboxyl-term (Appendix A). Conversely, the pose of NAD^+^ and Mn^2+^ reached stability after 20 ns, displaying an RMSD at or below 2 Å throughout simulation. Furthermore, putative MAL binding-site residues, based on previous reports, namely TYR85, ASP86, LYS156, ASP251, and ASP250 within MLE displayed movement of less than 2 Å (Figure 7B). Then, MAL was oriented through molecular docking simulations (Figure 7B).

### 2.5. Molecular Docking of Substrates of Malolactic Enzyme

Additionally, we evaluated the participation of the divalent cation on MLE mechanism by quantum polarized ligand docking (QPLD). Figure 8 illustrates the pose adopted by MAL inside the binding site of MLE. Malic acid interacts with MLE through coordination bonds with Mn^2+^, one LYS protonated residue, and several ASP residues interacting through hydrogen bonds. MAL-MLE interacting residues on this pose correspond with equivalent residues proposed for divalent-cation-dependent MAL decarboxylation.

We also calculated the most probable protonation state of MAL using the Epik module of the Schrodinger Suite. Results confirmed that MAL^2−^ is the most probable protonation state and thus interacting residues could be oriented differently to MAL^-^ and MAL.

Quantum polarized ligand docking (QPLD) was then employed to explore MAL^2−^ pose and binding energy (∆G_binding_). Results showed that all malic acid protonation forms lie in the same binding cavity, sharing the same set of binding amino acid residues (Figure 8). The latter interact mainly by hydrogen bonding and hydrophobic interactions (Appendix A). MAL^2−^ has the lowest ∆G_binding_, followed by MAL^−^ and MAL, with values of −23.8, −19.6, and −14.6 kJ/mol, respectively (Table 2). Interestingly, MAL^2−^ displayed an extended conformation, when compared with the other protonation states. This open conformation better satisfies the geometrical requirements of Mn^2+^ coordination geometry and the mechanism described for malic enzymes [29].

Geometrical stability of MAL inside the binding site was assessed after 200 ns molecular dynamics simulations of the MLE/MAL/NAD^+^/Mn^2+^ system. Of note, MAL does not remain on the site and exits the pocket at 25 ns. On the contrary, MAL^−^ and MAL^2−^ remain into the binding pocket throughout the whole simulation. Moreover, the average binding energy of the molecular interactions through the MM/GBSA rescoring method was calculated as it is relatively more accurate compared to single-structure theoretical determinations. MM-GBSA binding energies for MAL^−^ and MAL^2−^ showed binding affinity differences consistent with values from the ITC and QLPD experiments (Table 1 and Table 2, respectively). Furthermore, energy decomposition of MAL^−^ and MAL^2−^ interactions within the MLE binding pocket allowed to identify that binding is mainly driven by the negative charge interactions of the MAL carboxyl group with the positive charge of the side chain-N of LYS156, (Figure 7); whereas MAL^1−^ interacts with ASP86 and ASN396 mainly by H-bonding; and MAL^2−^ with TYR85, ASP86, ASP228, ASN396, and ASN440 mainly through water bridges.

## 3. Discussion

*In silico* analysis, MLE homology model together with QPLD and ITC experiments were carried out in the present study to determine the protonation state of l-malate required for its efficient decarboxylation to l-lactate by the malolactic enzyme of *Oenococcus oeni*.

The phylogenetic analysis showed that the evolution of the *O. oeni* malolactic enzyme is halfway between malic-malolactic enzymes of the genera Lactobacillus*-Pediococcus-Leuconostoc* and *Streptococcus-Lactococcus-Enterococcus*. The grouping of lactic acid bacteria in two clusters was in line with Makarova and Koonin (2007) [30], which employed the sequence of ribosomal proteins and RNA polymerase subunits for their phylogenetic analysis. Our results indicated that *O. oeni MLE* is part of a monophyletic group, together with the branch *Streptococcus-Lactococcus-Enterococcus,* whereas *O. Oeni MLE* constitutes a paraphyletic group of the *Lactobacillus-Pediococcus* branch. These results point that the genera *Streptococcus, Lactococcus, Enterococcus,* and *Oenococcus* descended from a common evolutionary ancestor, whose malolactic enzyme could share similar structural characteristics.

Malolactic fermentation usually occurs at pH levels between the range of 3.2 and 3.5, allowing for a rise in pH as the malic acid is converted to lactic acid. At this pH, the MAL^−^ protonated l-malic acid form prevails (Figure 2). On the other hand, several LAB, including *O. oeni* have an intracellular pH ≈ 5.0 [31,32], a condition where MAL^2−^ is the predominant protonated form. Several studies have reported that the malolactic fermentation reaction occurs in the *Oenococcus oeni* intracellular space, where pH is between 5.8 and 6.1; while in the extracellular medium, i.e., the wine conditions, the pH is within a range of 3 to 4 [33,34]. Additionally, Schümann et al. (2013) reported that *O. oeni* MLE has an optimum activity at pH 6.0 and 45 °C [14]. Accordingly, we evaluated the effect of pH on l-malic acid interaction with the MLE active site. To this end, we measured the enthalpy of reaction of *O. oeni* MLE with both cofactors, titrated with MAL at pH 4.5 and 5.3, using ITC. Our results showed a higher binding affinity for MAL^2−^ than for MAL^−^, in agreement with Schümann et al. (2013) [14]. Under the *O. oeni* intracellular conditions, the presence of some residues, in the binding pocket or in its vicinity, could accept the proton from the MAL^−^ form, predominant in solution, to lead the most stable MAL^2−^, such as Asp86 and Glu227 (see Figure 8A).

Although ITC experiments showed that pH significantly influences malic acid binding to *O. oeni* MLE, this method did not allow to extract structural information of the binding sites or enzymatic mechanisms, at atomic level. However, non-integer stoichiometric values indicated the formation or aggregation of dimers of higher quaternary structures, which was also observed by Dynamic Light Scattering measurements (data not shown). These results agree with the work of Schümann et al. (2012), where MLE of *O. oeni* was presented as a dimeric macromolecule, with each subunit having a functional binding site [35].

To give further structural insights and to understand the calorimetric results, we relied on the use of molecular docking and molecular dynamics methods. To this end, *O. oeni* MLE homology model was carried out, because the three-dimensional structure was not available. It is worth noting that SSN and phylogenetic analysis showed a close relation between malolactic and malic enzymes (Figure 4), although only four crystal structures are available as possible structural templates. Among these structures, malic enzyme from pigeon liver (PDB entry 1GQ2) and *E. coli* Malic enzyme (PDB entry 6AGS) were used as model templates, using the alignment shown in Figure 6. Both structures were identified as the closest related sequences according to the SSN; nevertheless, experimental data regarding 6AGS crystal is scarce, and was only used when no structural data from 1GQ2 were available. It is worth noting that NAD^+^, Mn^2+^ were incorporated as model constraints, using the pose of cofactors found on the 1GQ2 crystal. The MLE model was submitted to 200 ns molecular dynamics simulations to further explore its dynamic and stability and to identify relevant residues for malic acid binding. The trajectory analysis showed that, overall, residues near cofactors have slow mobility and form cavities that are suitable to bind l-malic acid.

Interestingly, SSN and phylogenetic analysis suggest a closed relation between malate dehydrogenases, malic enzymes, and malolactic enzymes; however, their binding sites are not completely conserved and thus, substrate pose, and binding residues could be oriented differently. To correctly orientate malic acid and to calculate theoretical binding energies considering Mn^2+^, we opted for the quantum polarized ligand docking (QPLD) method as it allows to properly calculate binding energies of metal-containing systems as it considers metal coordination, electronic polarization effect, among other missed terms in molecular-mechanics force fields. As can be seen in Table 2, QPLD results shown that MAL^−^ and MAL^2−^ have docking scores (∆G_binding_) that correlate with ITC measurement results. Although both MAL protonation states bind into the same cavity and share a set of amino acids, MAL^2−^ adopts an extended conformation, supported by hydrogen bonding and coordination geometry with Mn^2+^ (Figure 7). Further, we explore the binding site dynamics of the MAL, MAL^−^, and MAL^2−^ containing systems through molecular dynamics simulation. Of note, MAL^2−^ kept its orientation, as determined by the QLPD method, after 200 ns of molecular dynamics simulation, while MAL^−^ and MAL have more mobility inside the binding site. Furthermore, molecular dynamics provides a conformational ensemble that allows to calculate the average binding energy of the molecular interactions through the MM/GBSA rescoring method that is more accurate compared to single-structure theoretical determinations as it includes solvent effects. According to MM/GBSA results and ITC experiments, MAL^2−^ have the lowest binding energy (∆G_binding_) and the major energetic contribution is the stabilization of the two carboxylic group charges that interacts with Mn^2+^. Regarding ITC correlation with our molecular dynamics results, it should be noted that docking and MMGBSA calculations represent one of the steps of the reaction coordinates, that is pre-transition states without consider the diffusion pathways into the binding sites and omitting desolvation and other effects that directly impact the entropy variations observed by ITC measurements. This MAL^2−^ pose is in accordance with the mechanism described by Schümann et al. (2013), where Mn^2+^ act as an activator of the enzymatic catalysis and coordinate chemical reaction, while NAD^+^ act as oxidizing agent for oxidation of l-malate.

## 4. Materials and Methods

### 4.1. Analysis of Sequences and Construction of Phylogenetic Tree

The amino acid sequence of malolactic enzyme (MLE) from *Oenococcus Oeni* was retrieved from NCBI (accession number WP_002823502.1). Homologous amino acid sequences were found using the online available version of Basic Local Alignment Tools (BLAST™) [36]. Twenty five sequences (accession number EEV40786.1, AMG48999.1, BAQ56789.1, AOO75947.1, BAP84672.1, ALJ31288.1, ANZ58780.1, ALO02977.1, CCC78515.1, ALG26877.1, CAI54742.1, AOO73522.1, ARE28303.1, KLK96319.1, API72025.1, ANZ71056.1, AMV69835.1, ABJ68638.1, AQP43157.1, ABV10389.1, CCF03237.1, CBY99983.1, AEH55110.1, AEF25979.1, and ARC49389.1) of lactic acid bacteria were selected based on the e-value, the query coverage and its sequence identity with MLE, and were aligned using CLUSTAL OMEGA program (EMBL) [27]. The phylogenetic tree was built up using Interactive Tree Of Life [37].

To perform the Sequence similarity Network, the MLE sequence from *O. oeni* (Uniprot ID: Q48796) was aligned to the closest sequences (>70% id) using all-by-all BLAST within InterProScan database performed by the web service EFI-Enzyme Similarity Tool [38,39]. Each sequence was labeled by its primary biological function and structural data availability as provided by UNIPROT. Even though all the function entries were cured manually, the lack of consistency of UNIPROT terminology could lead to ambiguous descriptions. Finally, 309,755 sequences were admitted to the SSN building.

### 4.2. Protein Modeling

#### 4.2.1. Template Selection

The amino acid sequences of the malolactic enzyme from *O. oeni* strain DSM 20255 were retrieved from the NCBI (accession number ACX50963). The template was selected based on the e-value of the BLAST search, query coverage, and its sequence identity with MLE. Based on these criteria, malic enzyme from pigeon liver (PDB entry 1GQ2) was selected as a template to model MLE.

#### 4.2.2. Modeling of Malolactic Enzyme

A comparative model for the malolactic enzyme was constructed using Prime from Schrodinger Suite 2019 using the PDB 1GQ2 and 6AGS as a template. Both enzyme cofactors, NAD^+^ and Mn^+2^, were incorporated into the resulting models keeping the atomic coordinates from reference structure 1GQ2. The resulting model of the malolactic enzyme, including NAD^+^ cofactor and Mn^+2^ ion, was inserted in a water box and further neutralized with counter ions. Then MLE/NAD^+^/Mn^+2^ complex was subjected to cycles of energy minimize as described elsewhere and equilibration for 200 nanoseconds under NPT conditions.

#### 4.2.3. Ligand Preparation

l-malic acid three-dimensional structure was obtained from the PubChem database (Pubmed CID 222656) and prepared in Maestro (Schrödinger, LLC, New York, NY, USA) using the OPLS_2005 force field with default setting of the LigPrep package from Schrödinger. All molecules were visualized and pKa values were calculated using Epik at desire pH [40].

#### 4.2.4. Quantum Polarized Ligand Docking (QPLD)

l-malic acid and its other protonation states were docked with improved docking program of quantum polarized ligand docking (QPLD) of the Schrodinger Suite 2019 [41]. The best poses obtained by flexible ligand docking using Glide [42]. Then QM calculations were done using Jaguar to calculate the partial charges were replaced on the ligand in the field of receptor for each ligand complex [43]. Single point electrostatic calculations were carried out with the 6-31G*/LACVP* base set and B3LYP density functional theory, using the “Ultrafine” SCF accuracy level (iacc = 1, iac-scf = 2) for the QM region. Finally, ligand was redocked with updated atomic charges with the help of Glide XP and QPLD of the Schrodinger Suite 2019.

#### 4.2.5. Molecular Dynamics Simulation (MD)

MD simulations of malolactic enzyme and ligand complexes were carried out using Desmond and OPLS 2005 force field [44]. The protein ligand complexes were solvated with TIP3 water molecules. Sodium counter ions were added to balance the system net charge. The systems were submitted to the default Desmond protocol, which contains a series of restrained minimizations and MD simulations. The minimized system was relaxed under NPT ensemble for 50 ns equilibration simulation period, and 150 ns production simulations were carried out. Long range electrostatic interactions were computed by particle-mesh Ewald method and van der waals (VDW) cut-off was set to 9 Å.

### 4.3. Cloning and Expression of Recombinant Malolactic Enzyme

#### 4.3.1. Microorganisms, Plasmids, and Media

*Oenococcus oeni* [45,46] (PSU-1, ATCC^®^ BAA-331™) was obtained from the American Type Culture Collection (ATCC) (Virginia, USA). Cryogenically preserved (−80 °C) strains were cultured and maintained on MRS plates (Man, Rogosa and Sharpe) [47] and stored at 4 °C.

*Escherichia coli* BL21 strain and plasmid pET28a were obtained from Novagen (Buenos Aires, Argentina). Transformants were grown at 37 °C in LB medium, with the addition of 50 µg/mL kanamycin. Agar plates were made of LB media, including 15 g/l agar.

#### 4.3.2. Construction of the MLE Expression Vector

The malolactic enzyme gene was PCR amplified using genomic DNA from *O. oeni* strain PSU-1, extracted using the Wizard Genomic DNA purification kit (Promega). The 26 nt primers used for this amplification: 5’-GATATACCATGGGCAGCAGCATGACAGATCCAGTAAGTATTTTAAATGA-3 (forward) and 5´-CAGTGGTGGTGGTGGTGGTGGTATTTCGGCTCCCACC-3 (reverse), were designed based on the sequence of OEOE_RS07545 gene (1626 bp, NCBI). The linearized vector pET28a (5369 pb) was PCR amplified using the following 26 nt oligonucleotides: FWD 5´-ATACTTACTGGATCTGTCATGCTGCTGCCCATGG-3´; and 5´-TGAGGTGGGAGCCGAAATACCACCACCACCACCAC-3. Both pairs of primers were designed using SnapGene^®^ software (Chicago, IL, USA), to be employed for Gibson Assembly.

All PCRs to amplify DNA fragments suitable for Gibson assembly were carried out in 35 PCR cycles, using Phusion High-Fidelity DNA polymerase (Thermo Scientific, Waltham, MA, USA), following the manufacturer’s instructions. Gibson assembly was performed as previously described [48] with pairs of primers for each fragment to be assembled containing segments of about ∼40 bp homologous to the adjacent fragment to be linked. All PCR products were treated with the DpnI enzyme to eliminate original vector residues and purified by gel extraction using the Qiaquick Gel Extraction kit from Qiagen, according to the manufacturer’s instructions. The purified genes fragments and vectors were mixed based on their molar ratios in a final volume of 5 µL, containing 100 ng of total DNA. This DNA mix was added to 15 µL of 1.33X master mix (5X isothermal mix buffer, T5 exonuclease 1 U/µL, Phusion DNA polymerase 2 U/µL, Taq DNA ligase 40 U/µL and Milli-Q purified water), and the reaction mixture was incubated at 50 °C for 1 h. Finally, 10 µL reaction mix were used directly to transform chemically competent *E. coli* BL21 (DE3).

The vector construct, designated pET21a-MLE, was verified by sequencing (Macrogen Inc., Seoul, Korea). The resulting map is shown in Appendix A.

#### 4.3.3. Expression and Purification of Recombinant Malolactic Enzyme of *O. oeni*

*E. coli* BL21 (DE3) cells transformed with the pET21mle plasmid were grown at 37 °C and 140 rpm in 1 L shake flasks, containing 250 mL LB medium with 50 µL kanamycin. After 12 h incubation, MLE induction was performed by adding isopropyl β-d-1-thiogalactopyranoside (IPTG) to a final concentration of 0.5 mM. The cultures were incubated for another 12 h at 16 °C and 100 rpm. The resulting biomass was recovered from the fermentation broth by centrifugation (4000x g, 10 min, 4 °C) and the supernatant was discarded. Approximately 9 g of biomass were recovered from 1 L of fermentation broth. Subsequent cell disintegration was carried out in lysis buffer (Tris 20 mM pH 6.0, with 500 mM NaCl, 30 mM imidazole, and protease inhibitor cocktail complete™), at a concentration of 1 g of biomass in 10 mL of lysis buffer. The mix was distributed in 1.5 mL Eppendorf tubes with 250 μL of glass beads (Sigma-Aldrich^®^), and cell disruption was performed by agitation, three consecutive cycles of 30 s.

The crude extract was loaded onto immobilized metal affinity chromatography columns (HisTrap HP, 5 mL, Amersham Biosciences), operated with a peristaltic pump (with a flux 5 mL·m^−1^), and pre-equilibrated with binding buffer (HEPES 100 mM, KCl 100 mM, imidazol 20 mM, pH 6.0). The column was washed with 30 mL of binding buffer. The protein was eluted with 30 mL of stripping buffer (HEPES 100 mM, KCl 100 mM, imidazol 500 mM, pH 6.0), collecting fractions of 10 mL. The active fractions were pooled, desalted, and lyophilized. For experimental purposes, the protein was resuspended in HEPES buffer (100 mM HEPES, 0.5 mM NAD^+^, and 0.1 mM Mn^2+^, pH 6.0).

### 4.4. Calorimetric Characterization

Enthalpy changes associated with MLE-substrate interactions were measured using a Nano ITC instrument (TA Instruments Ltd., Crawley, West Sussex, U.K.), at 25 °C. An amount of 170 μL of MLE solution (30 μM, HEPES buffer at desired pH) were placed in the sample cell of the calorimeter and buffered substrate solution (100 μM, HEPES buffer at desired pH) was loaded into the injection syringe. The substrates were titrated into the sample cell as a sequence of 20 injections of 2.5 μL aliquots. The time delay (to allow equilibration) between successive injections was 3 min. The contents of the sample cell were stirred throughout the experiment at 200 rpm to ensure thorough mixing. Raw data were obtained as a plot of heat (μJ) against injection number and featured a series of peaks for each injection. These raw data peaks were transformed using the instrument software Nano Analyze (version 3.11.0, TA Instruments, New Castle, DE, USA) to obtain a plot of observed enthalpy change per mole of injectant against molar ratio and were corrected by subtracting the mixing enthalpies of the substrate solutions into protein-free solution.

## 5. Conclusions

In conclusion, in this work, we constructed a comparative model for MLE using the 3D structures of the malic enzyme from pigeon liver (PDB entry 1GQ2) and malic enzyme from *E. coli* (PDB entry 6AGS). Malic acid interactions within MLE binding pocket are mainly driven by hydrogen bonding and coordination with Mn^2+^, both dependent on the protonation state of the substrate. Our experimental and theoretical studies demonstrated that MAL^2−^ stabilizes the pose that fulfills the geometrical requirements to favor the malic acid decarboxylation catalyzed by MLE. Further theoretical and experimental studies are currently underway to provide more detailed information about the contribution of each residue on the MLE proposed mechanism.

## Figures and Tables

**Figure 1 molecules-25-03431-f001:**
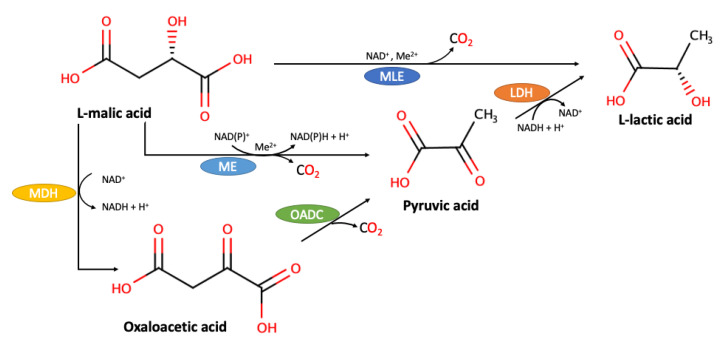
Possible decarboxylation pathways for the enzymatic conversion of l-malic acid to l-lactic acid. MDH, malate dehydrogenase; ME, malic enzyme; MLE, malolactic enzyme; OADC, oxaloacetate decarboxylase; LDH, lactate dehydrogenase. Adapted from Schümann et al. (2013) [14].

**Figure 2 molecules-25-03431-f002:**
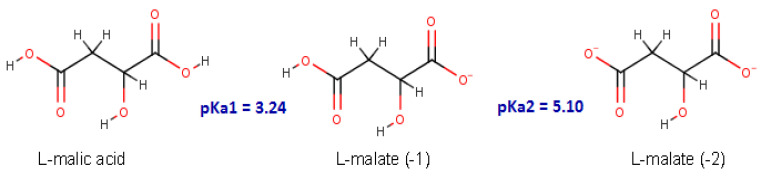
Chemical structure of l-malic acid and its protonation states.

**Figure 3 molecules-25-03431-f003:**
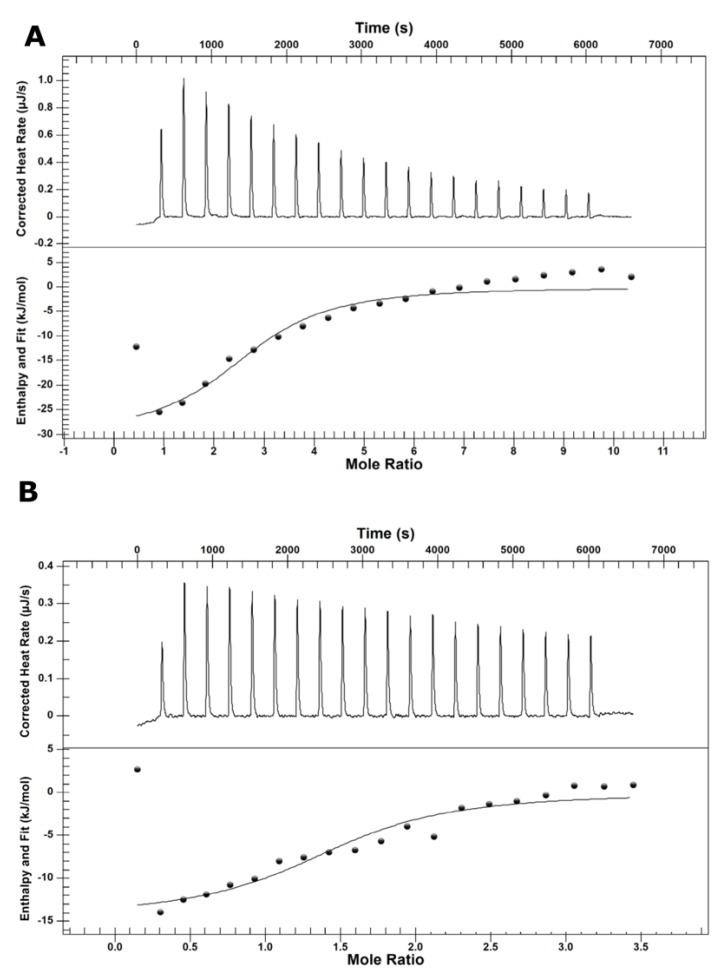
Binding isotherm curves of experimental calorimetric titrations of 0.3 mM l-malate (−2) (**A**) and (−1) (**B**) protonation states. Reaction was carried out by adding 30 mM malolactic enzyme to the reaction medium.

**Figure 4 molecules-25-03431-f004:**
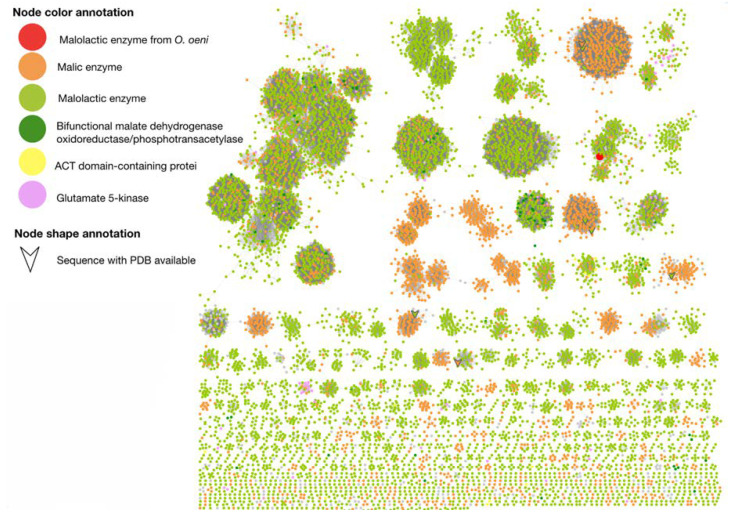
Sequence Similarity Network (SSN) of potential homologs to MLE of *O. oeni* with at least 70% identity of sequences. The nodes represent proteins and edges indicate similarity in amino acid sequence. Clustering by sequence identity is done with CD-HIT program. At values of sequence identity >70%, the nodes should contain sequences that share the same function; however, at lower values of sequence identity, the nodes may be functionally heterogeneous.

**Figure 5 molecules-25-03431-f005:**
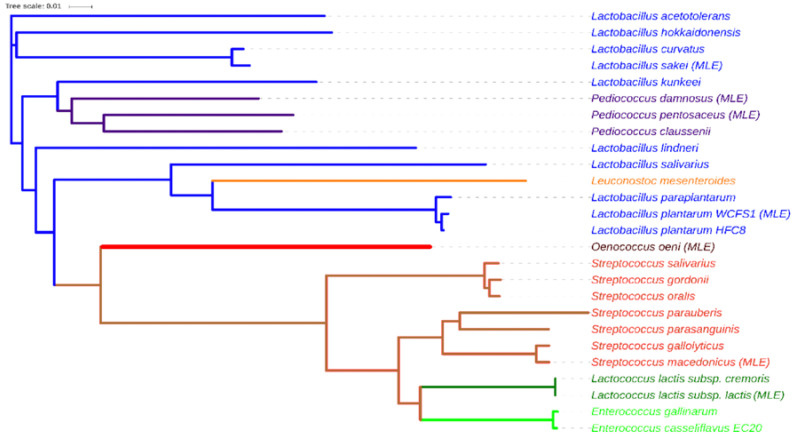
Phylogenetic tree of *Lactobacillales* constructed on the basis of multiple alignments of homologous sequences to *O. oeni* MLE, determined by Blastp. The multiple alignments and neighbor-joining tree were built using CLUSTAL OMEGA, and the visualization of the tree was done in iTOL.

**Figure 6 molecules-25-03431-f006:**
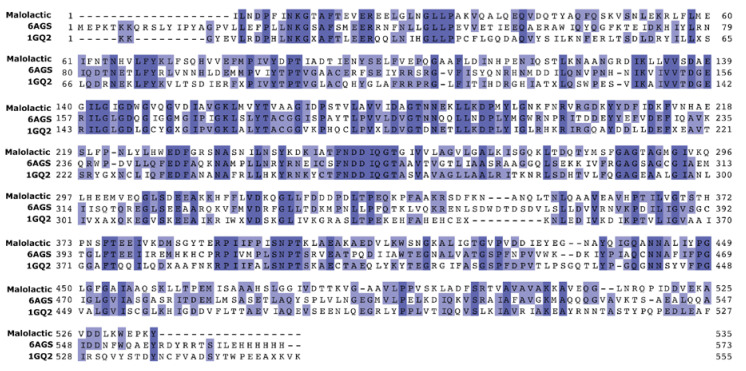
Sequence alignment of malolactic enzyme from *Oenoccocus Oeni*, malic enzyme from pigeon liver (PDB entry 1GQ2) and malic enzyme from *E. coli* (PDB entry 6AGS).

**Figure 7 molecules-25-03431-f007:**
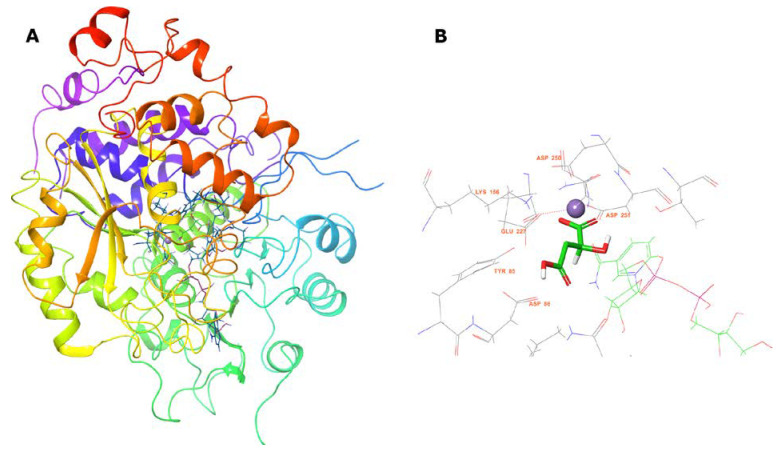
Homology model of *Oenoccocus oeni* MLE. (**A**) Protein structure after 200 ns MD simulation. (**B**) MAL pose inside predicted MLE binding site.

**Figure 8 molecules-25-03431-f008:**
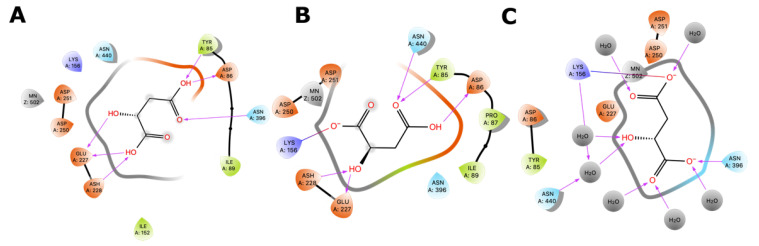
(**A**) l-Malic Acid and (**B**) l-Malate −1 and (**C**) l-Malate −2 pose into the MLE binding site predicted by the QLPD method.

**Table 1 molecules-25-03431-t001:** Binding energies (ΔG) of l-malate with malolactic enzyme using isothermal titration calorimetry.

Ligand	Kd (M)	n	∆G (kJ/mol)	*∆H* (kJ/mol)	*T∆S* (kJ/mol)
MAL^−^	3.19 × 10^−6^	2.7	−31.3	−30.0	1.3
MAL^2−^	1.29 × 10^−6^	1.5	−33.7	−14.5	19.2

Binding energies was calculated using an independent site interaction model. HEPES buffer (100 mM) was used to control pH and malic acid protonation form. Kd is dissociation constant, n correspond to non-integer stoichiometric values, Kd is the dissociation constant and ΔG is calculated as enthalpy (*ΔH*) minus *TΔS*.

**Table 2 molecules-25-03431-t002:** MAL^−^ and MAL^2^^−^ interactions with malic enzyme through 200 ns simulations.

MAL^−^	MAL^2−^
QPLD ∆G _binding_ −19.6 kJ/mol	QPLD ∆G_binding_ −23.8 kJ/mol
MM/GBSA ∆G _binding_ −154.8 kJ/mol	MM/GBSA ∆G _binding_ −175.7 kJ/mol
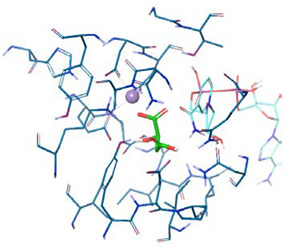	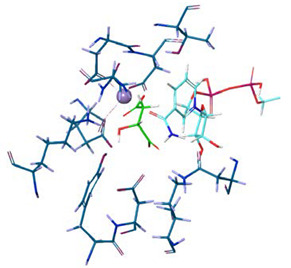
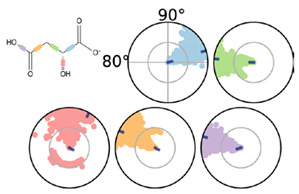	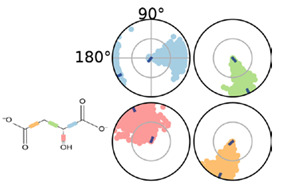
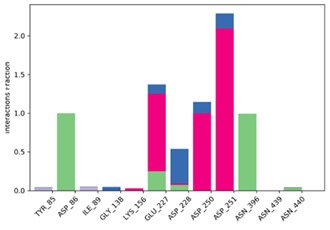	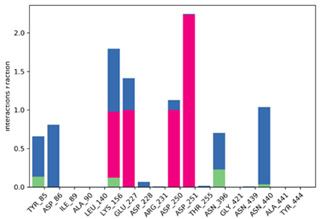

First row depicts MAL^−^ and MAL^2−^ representative conformations. Second row describes the conformation of the torsion throughout the course of the simulation. The beginning of the simulation is in the center of the radial plot and the time evolution is plotted radially outwards. Third row describes the kind of interactions both of MAL^1^^−^ and MAL^2^^−^ with amino acids of binding pocket, stacked bar charts are normalized over the course of the trajectory. Green represents H-bond, purple hydrophobic contacts, magenta ionic contacts and blue water bridges. Only the last 150 ns were used for calculations.

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
