# Peer review of "l-Malate (−2) Protonation State is Required for Efficient Decarboxylation to l-Lactate by the Malolactic Enzyme of Oenococcus oeni"

_molecules, 2020, doi:10.3390/molecules25153431_

Round 1
Reviewer 1 Report
The authors investigate one of the possible mechanisms of malate decarboxylation into lactate, as it is presumably done by O. oeni via catalysis of malolactic enzyme (MLE, the sentence at page 2 line 58 is a little confusing).
The authors measured binding free energy of malate to MLE by ITC.
They found by genetic trees and homology models candidate enzymes for describing, at atomistic level, the MLE/malate enzyme/substrate (ES) complex that is supposed to be the intermediate in the investigated mechanism.
According to sequence similarity, the authors used 1GQ2 and 6AGS in PDB as template structures to perform short MD simulations and docking studies.
They performed simulations including NAD^+ and Mn^2+ essential cofactors, but the force field for these components is not described.
The metal cation is particularly important for the described results, since electrostatics is found particularly relevant. The small difference in binding free energy can be dramatically influenced
by the model of this cation, since the details of coordination chemistry are likely not included in the force field.
They found the MLE/Mal^2- as the most stable ES complex. This result is of relevance at pH>5.
Hydrogen bonds (see possible contributions from tautomers) and electrostatics are the major contribution to the stabilization of ES complex.
This work is a useful structural information additional to that of previous works on the subject, like ref. 33 in the manuscript.
The efficiency of the proposed mechanism at pH around 6 (where malate is dominant over protonated forms) does not appear very useful for industrial application, where pH is constrained
below 4 (see refs. cited in ref. 33 in the manuscript).
Therefore, some perspectives on further work on the subject are expected in this manuscript.There are some possible issues about the decarboxylation of malate as later step in the mechanism of lactate formation (Fig.1 in ref. 14 in this manuscript): is there any possible hint on the next step once the representative ES structure (Table 2, right panel) is given? A tentative answer is expected by the reader.
Fig.1 - There is no mention of tautomers of malate. It would be interesting to discuss the contribution (if any) of tautomers to acidity as well as to MLE binding.
Section 2.2 and Fig.3 - The network can not be understood, with the information provided, by someone not expert in the field. Please, explain better the meaning of dots etc..
The reader can understand that this is a method to obtain the structure most suitable as a starting
point for atomistic modeling, but the section is not clear.
ITC data discussed at line 233 (higher affinity for MAL^2- at high pH) are not displayed.
Section 4.2.2, line 311 - REF is missing; see above for missing description of the force field used in modeling.
Some methods (Epik, LigPrip, QPLD, all modules of Schroedinger Inc. package) deserve at least
a reference in available scientific literature.
Reproduction of results should be made possible, even with no access to Schroedinger Inc. package.
MAL^-2 is usually MAL^2-
MAL^-1 -> MAL^-
Reviewer 2 Report
The manuscript reports an interesting combined theoretical and experimental study aimed at clarifying the mechanisms of decarboxylation enzymes in wine making. Particular emphasis is put in elucidating the functioning of malolactic enzyme and the protonation state of malate acting as a substrate.
The study is interesting and the alliance between phylogenetic approaches, enzyme expression and calorimetric determination and molecular modeling particularly convincing, especially for the determination of structures by homology model.
However, some scientific issues should be addrezssed by the authors before acceptation of the contribution.
In particular the enzyme is operating in acid conditions ph (3.2-3.5) have the authors taken this into account when considering the protonation state of the protein residues. This should be precised and carefully considered. Also it would be important to discuss the presence, in the binding pocket or in its vicinity, of some residues that could accept the proton from the MAD-1 form, predominant in solution, to lead the most stable MAD-2. For example I noticed the presence of a glutamate. This aspect should be checked, they have all the data to do that, and properly discussed by the authors.
Other minor concerns
in the introduction instead of 'optimal conditions' I would use 'required optimal conditions'
It would be useful to add schemes with the chemical reactions of the three proposed mechanisms. This could be easily integrated into Figure 1
There is probably a missing word in the sentence 'After purification of the, we measured'
Form Table 1 one assumes a huge effect of the entropic factor, could the author comment on this ? Also how it correlates with the molecular simulations results ?
The authors says that the neutral MAL rpidly exits the pocket, could they quantify in terms of nanoseconds ?
By decomposing MM/GBSA energy the authors identify the main interacting mainoacids, it would be useful to provide a time series of the evolution of the distances of the proposed hydrogen -bond or salt bridges over the MD trajectory.
The results of the binding energies in Table 2 should be provided in kJ/mol to be directly comparable with the experimental ones. Also I would dissociate Table and Figure.
Reviewer 3 Report
Review
------
The authors investigate the effect of the protonation state of L-malic acid on its binding affinity to the malolactic enzyme (MAL) in Oenococcus oeni through quantum polarized ligand docking to a MAL homology model.
The binding free energy values they obtain for MAL, MAL⁻¹, and MAL⁻² are consistent with calorimetry measurements and explain why MAL is most active in the 3.0 to 6.0 pH range.
The authors have done a comprehensive job on both the experimental and computational aspects of this work and provide quite a bit of evidence to support one particular mechanisim of malolactic fermentation.
The materials and methods section provides enough information to reproduce the computational aspects of this work.
I recommend publication after a few minor points are added:
1. Could the authors place the low sequence identity between 1GQ2 and MLE in context by citing other homology model-enabled studies with similarly low sequence identity and coverage?
2. Could the authors add error estimtes for their QPLD and MM/GBSA ΔG estimates?
Typos:
pg 2, L46: Should "< 50 ppm" be "> 50 ppm" (for _high_ sulfite concentration)?
Round 2
Reviewer 2 Report
The authors have satisfactorily answered to all my concerns.
I fully support pubblication of this interesting and well conducted study.